# DOMAIN SHIFT TUNING OVER KNOWLEDGE GAP

## ABSTRACT

This paper introduces Domain Shift Tuning (DST), a novel framework designed to guide pre-trained language models (PLMs), including Large Language Models (LLMs), in overcoming domain discrepancies (i.e., source-target). PLMs, pre-trained on extensive and diverse corpora, the source domain, often encounter domain gaps after fine-tuning over the target domain. Unlike conventional adapters or Parameter-Efficient Fine-Tuning (PEFT) methods, DST conceptualizes domain gaps as differences in knowledge encapsulated within multiple subnetworks of PLMs. To bridge this gap, our challenge is to find a subnetwork set that corresponds to these pieces of knowledge and their weight. This direction leads DST to employ a lightweight subnetwork, the Knowledge Steering Layer (KSL), and a training objective, Knowledge Distribution Modeling (KDM). These components enable DST to fine-tune PLMs by aligning the knowledge weights of the source domain with those of the target domain. Experimental results on diverse datasets demonstrate that DST effectively mitigates the domain gap, allowing PLMs to generate text that closely aligns with even a small target corpus, thereby significantly enhancing domain adaptation for PLMs at lower computational cost.

## 1 INTRODUCTION

PLMs, including LLMs, have demonstrated a powerful capability to generate high-quality text. However, their effectiveness is often limited by the size of the target corpus, which is typically much smaller than the source corpora used for training PLMs. For instance, the popular pre-training datasets of Giga5en Parker et al. (2011), and ClueWeb 2012-B[1] occupy 16G, and 25TB, respectively. This size discrepancy can lead to catastrophic forgetting and poor generalization Lin et al. (2023), especially when all weights of the PLMs are fine-tuned. Given the swift diversification of PLM applications, techniques are needed that can effectively achieve domain adaptation Malik et al. (2023); Diao et al. (2023); Zhang et al. (2024) or PEFTs Hu et al. (2022); Dettmers et al. (2023); Xu et al. (2023); Wu et al. (2024).

In response to this need, we propose a model-agnostic adaptation framework, Domain Shift Tuning (DST), to tune PLMs toward the target domain. DST is based on the idea that PLMs encapsulate multiple pieces of knowledge as subnetworks, with each domain represented as weights over these subnetworks. The domain gap is thus represented as the difference in weights over these subnetworks, specifically those unique to the source and target domains. To find knowledge-equivalent subnetworks in the PLM and infer their weights, DST introduces a lightweight subnetwork, the Knowledge Steering Layer (KSL), and a training objective, Knowledge Distribution Modeling (KDM). Unlike other adapters Wang et al. (2022) and PEFTs, DST's novelty lies in associating domains and PLMs using knowledge and tuning the PLM by finding knowledge-equivalent networks and adjusting their weights.

Our experiments confirm the effectiveness of DST, demonstrating its theoretical and practical contributions;
**Theoretical**: KSL provides a differentiable access mechanism to represent domain knowledge as weights over multiple subnetworks of the PLM and fine-tune the PLM by adjusting this weight.
**Practical**: The framework's model-agnostic nature allows it to be applied to various PLMs at lower computational cost, enhancing its versatility and compatibility with other adapters and PEFTs.

---

[1] https://www.lemurproject.org/clueweb09.php/

## 2 PREVIOUS WORK

Transformer Vaswani et al. (2017) based PLMs Devlin et al. (2019); Radford et al. (2019); Yang et al. (2019); Liu et al. (2019); Lan et al. (2020) have made significant strides in Natural Language Processing (NLP) tasks, excelling at exploring local token relationships over global semantics Wang et al. (2020). However, they face challenges in adapting to tasks that require domain shift using topics. This paper introduces DST, a novel approach to address these challenges. PLMs, including BertSUM Wang et al. (2020) and UNIfied pre-trained Language Model Dong et al. (2019), have shown promise in various NLP tasks. Despite their success, these models struggle to capture explicit document semantics as effectively as topic models Wang et al. (2020). DST aims to bridge this gap by adapting PLMs to tasks requiring domain shift using topics.

Continual pretraining Gururangan & et al (2020) has demonstrated the benefits of optimizing a PLM to a target domain before further fine-tuning. UDALM Karouzos et al. (2021) and AdaPrompt Chen et al. (2022) follow a similar approach, training PLMs on the target domain and then training a target classifier with source domain labeled data. Ke et al. (2022) decouple global and domain-specific knowledge through continual pre-training on a domain-specific corpus. DST separes PLMs into sub-networks using latent discrete variables, each representing global or local knowledge.

To mitigate catastrophic forgetting, PEFT methods have been introduced to keep most of the PLM weights frozen. While AdaMix Wang et al. (2022) can leverage a mixture of adapters like Houlsby et al. Houlsby et al. (2019), a mixture of low rank decomposition matrices like LoRA Hu et al. (2022) and a minimal additional parameters like $(IA)^3$ Liu et al. (2022) to improve downstream task performance while keeping most of the PLM weights frozen. While PEFT methods focus on reducing the number of fine-tuning parameters and memory usage, DST focuses on bridging the domain gap by adjusting subnetwork weights within PLMs.

Topic models Blei et al. (2003); Wang et al. (2020) and their extensions Dieng et al. (2016); Jo et al. (2017) take a global statistical view and look at the word distributions of topics across a given corpus. Although these models organize a given corpus into small sets of prominent topics and have been proven to be powerful tools for uncovering latent structure, they and their application Chang et al. (2021); Wang et al. (2018; 2020) are not, in the strict sense, sequence models. Xu et al. (2024) introduces Energy-Based Concept Bottleneck Models as a unified framework for concept-based prediction, concept correction, and fine-grained interpretations based on conditional probabilities.

Like other PEFT, DST updates only the weights over subnetworks on the top of PLM rather than overwriting entire parameters. As our introduced lightweight subnetwork is placed in a different location than the other adapters, DST can be compatible with training strategies, adaptation modules, prompt tuning Lester et al. (2021), or continuous prompt models Li & Liang (2021); Liu et al. (2021); Zhang et al. (2022).

## 3 OUR FRAMEWORK

### 3.1 MOTIVATION, CONCEPT, AND DEFINITIONS

DST aims to bridge the domain gap by leveraging disparities between global (e.g., domain-agnostic linguistic elements) and local (e.g., domain-specific semantic elements) knowledge, as depicted in Figure 1(left), automatically encapsulate these disparities about a given target dataset. The underlying hypothesis is that each piece of knowledge is embedded within a subnetwork of PLMs, akin to the lottery ticket hypothesis Frankle & Carbin (2019). Here, knowledge is considered a latent and relative concept, not as concretely defined as topics in topic models. PLMs encode knowledge as token-level embeddings and subnetworks, referred to as latent **knowledge**. As these knowledge disparities are relative and corpus-dependent, it is difficult to show a clear definition. Subnetwork weights include both the original PLM network weights and newly added network weights. While the PLM parameters remain frozen, the newly introduced parameters are updated to adjust these weights. PLMs, which are neural language models tailored for specific tasks such as text generation Bengio et al. (2003), encapsulate both global and local knowledge across source and target domains. Given a text sequence $\mathbf{x}_d = \{x_{d,1}, \cdots, x_{d,|x_d|}\}$ and dataset $D = \{\mathbf{x}_1, \cdots, \mathbf{x}_D\}$, PLMs are

Figure 1: (left) The domain gap between the source and the target can be shown by using global and local knowledge, and can be interpreted as the difference between the knowledge distributions, and token frequencies, where each knowledge has its token distributions. The arrows indicate the selection of token distribution from the knowledge distribution of global and local knowledge. These differences affect the embedding representation of each token and are stored as parameters of PLMs. (center) The architecture overview of DST: DST consists of a Knowledge Steering Layer (KSL) and Knowledge Distribution Modeling (KDM), where TID denotes the last hidden state and KDM takes TID as input. Without changing the structure of PLMs, both KSL and KDM are inserted between Transformer blocks and LM Head (LMH), which are their common components. (right) The detail of KSL with the affine in Eq (4). DST updates parameters used in the KSL, $\mathbf{W}_Z$, $\mathbf{W}_{az}$, and $\mathbf{b}_z$ using TID and KDM on each text while freezing other parameters in the PLM.

pre-trained by maximizing the likelihood under forward autoregressive factorization:

$$\mathcal{L}_{LM}(\theta) = \sum_{d=1}^{|D|} \sum_{t=1}^{|x_d|} \log P_\theta(x_{d,t}|\mathbf{x}_{d,1:t-1}), \tag{1}$$

where $\theta$ denotes model parameters, and $x_{d,t}$ represents the $t$-th token (word) in the $d$-th text.

The divergence between domains is reflected in token distributions and their domain-specific distributions, i.e., the knowledge distribution, as shown in Figure 1(left). For example, a generic PLM might predict "Michelangelo" as the next token for the sentence "My favorite artist is," while a fine-tuned PLM might suggest "Botticelli." This discrepancy leads to representing topics using latent variable indicators, rather than hidden spaces obtained in Variational Autoencoders (VAEs). To discretely represent the knowledge embedded in the PLM as subnetworks, we follow the concept of topics and introduce these indicators, denoted as $\mathbf{z} \in \mathbb{R}^K$, into the PLMs, modifying the likelihood function as follows:

$$\mathcal{L}_{MLM}(\theta) = \sum_{d=1}^{|D|} \sum_{t=1}^{|x_d|} \log \sum_{z_t=1}^{K} \underbrace{P_\theta(x_{d,t}|z_t, \mathbf{x}_{d,1:t-1})}_{\text{knowledge specific token distribution}} \underbrace{P_\theta(z_t|\mathbf{x}_{d,1:t-1})}_{\text{knowledge distribution}}, \tag{2}$$

where $z_t$ indicates the distribution used for the $t$-th token, knowledge, and $\mathbf{K}$ is the number of (latent) knowledge. The Mixture Language Model (MLM) partitions a PLM into knowledge-equivalent subnetworks via $\mathbf{z}$, $P_\theta(x_{d,t}|z_t, \mathbf{x}_{d,1:t-1})$, enhancing its ability to generate coherent and contextually relevant content. The integration of $\mathbf{z}$ aligns generated text with specific knowledge, and knowledge distribution. While BERTopic Grootendorst (2022) generates topic representations, and TopClus Meng et al. (2022) proposes a joint latent space learning and clustering framework, they overlook the disparities between these domains. Note that $P_\theta(z_t|\mathbf{x}_{d,1:t-1})$ is a multinomial distribution over discrete variables, unlike the Gaussian distribution used in variational autoencoders Kingma & Welling (2014) or its extensions Wang & Wan (2019); Zhu et al. (2021); Cai & Cai (2022). Similar to topic models, this knowledge is latent and inferred from data, requiring additional analysis or visualization for interpretation. Eq (2) ensures that the concept of partitioning PLMs into $K$-subnetworks, $P_\theta(x_{d,t}|z_t, \mathbf{x}_{d,1:t-1})$, via discrete variables $\mathbf{z}$ and adjusting their weights using knowledge distribution, $P_\theta(z_t|\mathbf{x}_{d,1:t-1})$, to align the source domain with the target domain. This paper follows Eq (2) and explores where to insert in the common architecture of PLMs, Transformer, and the framework to find subnetworks and their training objectives to fine-tune the PLM.

## 3.2 Architecture design

Motivated by previous works Li et al. (2018); Houlsby et al. (2019); Aghajanyan et al. (2021), LoRA Hu et al. (2022) injects trainable rank decomposition matrices into each layer of the Trans-

former architecture. Unlike LoRA, DST focuses on both the risks of overfitting and the global-local knowledge differences rather than exploring the lower intrinsic dimension of the source domain knowledge. Ramasesh et al. (2021) pointed out that catastrophic forgetting occurs mainly in the higher layers. It is often observed that the learned attentive patterns from many heads are not as reasonable as we expect Michel et al. (2019), and we might obtain this global information from the upper blocks by increasing the number of transformer blocks Dosovitskiy et al. (2021); unfortunately, as the transformer architecture requires a large number of parameters, its computational cost is very high. As shown in Figure 1(center), DST places a Knowledge Steering Layer (KSL) on the top of the Transformer layer and updates only its related parameters to avoid catastrophic forgetting. This does not break any PLM structure and allows the reuse of PLMs and their parameters. As 1) knowledge describes a co-occurrence pattern of tokens with similar semantics, and 2) the differences between the pre-training and the fine-tuning datasets are not only in the knowledge itself but also in the ratio of knowledge, we develop a training task, Knowledge Distribution Modeling (KDM), to align knowledge to each text. Since global distributions do not require additional learning, DST is designed to find target-specific distributions through knowledge, and update them, $P_\theta(z_t|\mathbf{x}_{d,1:t-1})$ and $P_\theta(x_{d,t}|z_t, \mathbf{x}_{d,1:t-1})$, in fine-tuning. As shown in Eq (2) and Figure 1(right), This design enables PLMs to interpret $P_\theta(z_t|\mathbf{x}_{d,1:t-1})$ as the distribution over knowledge and $P_\theta(x_{d,t}|z_t, \mathbf{x}_{d,1:t-1})$ as the distribution over next tokens, and emphasize local knowledge that might otherwise be buried, thus preventing catastrophic forgetting.

### 3.3 KNOWLEDGE STEERING LAYER (KSL)

Figure (1)(left) shows that each knowledge has each token distribution. This is the rationale behind the DST positioning the KSL atop the Transformer layers to guide the text decoder. The KSL transforms the hidden representation vector $\mathbf{H}_L = [h_{L,1}, \cdots, h_{L,|x|}] \in \mathbb{R}^{|x| \times d_h}$ into a indicator vector $\mathbf{z} \in \mathbb{R}^K$, subsequently selecting the knowledge-specific token distribution in each text. That is $\mathbf{z}$ identifies the learnable weights (matrix), not each knowledge itself.

This transformation results in Eq (2) by defining the knowledge matrix $\mathbf{W}_Z \in \mathbb{R}^{d_h \times K}$ and the token generation function $\mathcal{F}(\mathbf{h}_{L,t})$. These matrices are applied to $\mathbf{h}_{L,t} \in \mathbb{R}^{d_h}$ in the text decoder, yielding $\mathcal{X}_T$, which is utilized to sample the next token, $x_i$, as a verbalizer, according to the probability:

$$P_\theta(z_t|\mathbf{x}_{d,1:t-1}) \propto LayerNorm(\mathbf{h}_{L,t})\mathbf{W}_Z, \quad P_\theta(x_{d,t}|\mathbf{x}_{d,1:t-1}, z_t) \propto \mathcal{F}(\mathbf{h}_{L,t}, z_t),$$

$$P_\theta(x_{d,t}|\mathbf{x}_{d,1:t-1}) = \sum_{z_t=0}^{K} P_\theta(x_{d,t}|\mathbf{x}_{d,1:t-1}, z_t)P_\theta(z_t|\mathbf{x}_{d,1:t-1}), \quad x_i \sim P_\theta(x_{d,t}|\mathbf{x}_{d,1:t-1}) \tag{3}$$

where $\mathbf{W}_Z$ are learnable weights, and $x_i$ is the score of the $i$-th token in the vocabulary. As $P_\theta(x_{d,t}|\mathbf{x}_{d,1:t-1})$ provides the probability over tokens, the next token is chosen by sampling a multinomial distribution with probabilities clipped to the top-$k$ tokens. Regarding $\mathcal{F}(\mathbf{h}_{L,t}, z_t)$, we apply the language model head (LMH) to it. For the output of Transformer blocks, $\mathbf{h}_{L,t}$, we adhere to the conventional activation functions (e.g., addition, multiplication, and affine) and propose three transformations to produce $x_{d,t}$ that correspond to the given $z_t$ and $\mathbf{x}_{d,1:t-1}$, $\mathbf{h}_{L,t}$.

$$\mathcal{F}(\mathbf{h}_{L,t}, z_t) = LMH(\mathbf{h}_{L,t}), \quad \mathbf{h}_{L,t} = \begin{cases} \mathbf{h}_{L,t} & \text{residual if } z_t = 0 \\ \mathbf{h}_{L,t}\mathbf{W}_{az} + \mathbf{b}_{\mathbf{z}} & \text{affine if } z_t = z \text{ and } z > 0, \end{cases} \tag{4}$$

where $LMH()$ is the LM head, $\mathbf{g}_z \in \mathbb{R}^{d_h}$, $\mathbf{W}_{az} \in \mathbb{R}^{d_h \times d_h}$, and $\mathbf{b}_z \in \mathbb{R}^{d_h}$ are the $z$ specific learnable weights. We prepare the residual to select the input if $z = 0$, take $\mathbf{h}_{L,t}$ as the global token distribution, and partition the PLM into subnetworks shown in Figure 1, which preserves the PLM functionality, propose an alternative (i.e., addition $((1 - \omega)\mathbf{h}_{L,t} + \omega\mathbf{g}_z)$, multiplication $(\mathbf{h}_{L,t} \otimes \mathbf{g}_z)$, and affine) for $z > 0$, and confirm by an ablation analysis that affine is the best function. The subnetworks in the PLM differ only in $\mathbf{h}_{L,t}$, which is divided by KSL, and share the other networks.

While this subnetwork fine-tunes only a subset of PLM parameters like other techniques, it differs in following Eq (2) and explicitly incorporating the concept of discrete bayes. While Variational Autoencoder (VAE)-based models usually face the posterior collapse problem, KSL considers knowledge as a quantized sample of the underlying token distribution rather than conventional topic models, and samples latent index $z$ in each token just as the final layer of PLM samples the token. This setting ensures that DST can update knowledge-related parameters, including distribution through

training (i.e., backpropagation with cross-validation over tokens and training tasks) like other hidden variable parameters. The ratio of global token distributions and the nature of target-specific token distributions are both contingent on the provided target corpus. These are relative differences that become apparent post the fine-tuning or freezing of PLMs (i.e., freezing $LayerNorm(\mathbf{h}_{L,t})$ in Eq (3)).

Note that just as Eq (1) is transformed into Eq (2) through the introduction of $\mathbf{z}$, the top layer of previous Transformer-based PLMs is decomposed into the product of $\mathbf{W}_Z$ and $\mathcal{F}(\mathbf{h}_{L,t})$ in Eq (3). While both $\mathbf{W}_Z$ and $\mathbf{W}_{az}$ ($\mathbf{b}_z, \mathbf{g}_z$) are newly introduced parameters, other parameters in the Transformer blocks and LMH of PLM are frozen. Different from other PLMs, DST 1) aligns the $t+1$-th knowledge of target text, $P_\theta(z_t|\mathbf{x}_{d,1:t-1})$, and weights $P_\theta(x_{d,t}|\mathbf{x}_{d,1:t-1}, z_t)$ according to the distribution over $\mathbf{z}$, and 2) samples each token according to $p(x_i \in \mathcal{X}_T)$.

The top hidden state, $\mathbf{H}_L$, reflects the contextualized representation of the whole sequence in the decoder. As DST distills the target-specific knowledge via $\mathbf{z}$, the average of the token-level hidden states over each $i$-th text corresponds to a topic distribution of topic models, and N-gram topics by incorporating both the preceding topics and the topic-specific distributions over tokens. As is clear from Figure (1), KSL can be applied to the Transformer encoder framework (e.g., BERT) as well as the Transformer decoder framework (e.g., GPT-*, Llama-3). That is, we can modify of $P_\theta(z_t|\mathbf{x}_{d,1:t-1})$, $P_\theta(x_{d,t}|\mathbf{x}_{d,1:t-1}, z_t)$, $P_\theta(x_{d,t}|\mathbf{x}_{d,1:t-1})$, $P(x_i \in \mathcal{X}_T)$ in Eq (3) to $P_\theta(z_t|\mathbf{x}_{d,\neg t})$, $P_\theta(x_{d,t}|\mathbf{x}_{d,\neg t}, z_t)$, $P_\theta(x_{d,t}|\mathbf{x}_{d,\neg t})$ as follows:

$$P_\theta(z_t|\mathbf{x}_{d,\neg t}) \propto LayerNorm(\mathbf{h}_{L,t})\mathbf{W}_Z, \quad P_\theta(x_{d,t}|\mathbf{x}_{d,\neg t}, z_t) \propto \mathcal{F}(\mathbf{h}_{L,t}, z_t),$$

$$P_\theta(x_{d,t}|\mathbf{x}_{d,\neg t}) = \sum_{z_t=0}^{K} P_\theta(x_{d,t}|\mathbf{x}_{d,\neg t}, z_t) P_\theta(z_t|\mathbf{x}_{d,\neg t}), \quad x_i \sim P_\theta(x_{d,t}|\mathbf{x}_{d,1:t-1}), \tag{5}$$

where $\mathbf{x}_{d,\neg t}$ means $\mathbf{x}_d$ excluding $x_{d,t}$.

## 4 MODEL TRAINING

### 4.1 KNOWLEDGE DISTRIBUTION MODELING (KDM)

Inspired by the principles of contrastive learning Khosla et al. (2020) and triplet loss, the KDM is designed to minimize discrepancies between texts at both the knowledge and hidden representation levels, and use the distance between text pairs within the same batch. Models based on the Transformer encoder, such as BERT, employ a special token, [CLS], to encode an entire input and derive its representation. However, models based on the Transformer decoder do not have an equivalent token. To address this, we append [CLS] to the end of each input text for Transformer-decoder-based PLMs, as illustrated in Figure 1(center). This modification allows these PLMs to learn representations directly through $\mathbf{z}$ at the input-level representation, thereby obtaining a knowledge distribution $\mathbf{z}_d \in \mathbb{R}^K$ specific to the $d$-th text on the KSL. Given that texts with similar content are likely to share similar knowledge distributions, we can define the similarities between texts $SIM_z \in \mathbb{R}^{\mathbf{B} \times \mathbf{B}}$ using $\mathbf{z}_*$, where $\mathbf{B}$ represents each batch. As depicted in Figure (1)(center), $TID_d$ is the final output of the last token of the $d$-th input text sequence and serves to represent each text, much like [CLS] in BERT. We define another text similarity using $TID_*$ as $SIM_{TID} \in \mathbb{R}^{\mathbf{B} \times \mathbf{B}}$. Mathematically, this objective minimizes the following loss function:

$$\mathcal{L}_{KDM}(\theta) = \min_{(i,j)\sim\mathbf{B}} (||SIM_z - SIM_{TID}||),$$

$$SIM_{layer}(i,j) = \begin{cases} SIM_z(i,j) = \mathcal{F}_{sim}(\mathbf{z}_i, \mathbf{z}_j) & \text{if layer is KSL} \\ SIM_{TID}(i,j) = \mathcal{F}_{sim}(TID_i, TID_j) & \text{else} \end{cases}, \tag{6}$$

where $\mathcal{F}_{sim}(i,j)$ is the similarity function between the $i$-th and $j$-th text, and uses Kullback–Leibler divergence (upper) and a simple cosine function (lower).

### 4.2 TRAINING OBJECTIVE OF DST

We employ a unified multi-task learning framework. As DST can adapt PLMs, their parameters, $\theta$, of Eq (2) are used to initialize the KSL, and a fine-tuning process is used to adapt $\theta$ to the fine-tuning

Table 1: Basic statistics of the datasets used in this paper

| Task category | Dataset | #reviews | #vocabulary | K(#topics) |
|---|---|---|---|---|
| Topic discovery and text classification | NYT | 31,997 | 25,903 | 100 |
| Text generation | Amazon | 210,000 | 246,534 | 10,20,30 |
| | arXiv | 1,506,500 | 565,762 | 10,20,30 |

data. To optimize these parameters and bridge the gap between the data used in the pre-training and the fine-tuning process, we optimize the model loss in this tuning process. Using Eq (2),(6), we can define the loss function, $\mathcal{L}_{DST}(\theta)$, as the sum of these objective functions, and is to be optimized in the fine-tuning stage:

$$\mathcal{L}_{DST}(\theta) = -\mathcal{L}_{MLM}(\theta) + \lambda_{KDM}\mathcal{L}_{KDM}(\theta), \tag{7}$$

where $\theta$ is the parameter set of DST, $\lambda_{KDM}$ are hyper-parameters that balance the importance of MLM and KDM.

# 5 EXPERIMENTS

## 5.1 DATASETS AND EXPERIMENT DESIGN

**Datasets** We conducted evaluations using The New York Times annotated corpus (NYT)[2], Amazon review[3], and arXiv Dataset[4], as they are large publicly available datasets and can be manually evaluated by screened colleagues. The experiments focus on review texts (Amazon), news articles (NYT), and scientific articles (arXiv). Although including a broader range of domains, such as social media and legal texts, would better demonstrate DST's generalizability, we select these datasets since the resulting data size is computationally feasible on a general-purpose server, includes a variety of topics that are different from the pre-training corpus, meets the public reproducibility requirement, and can validate our insight that a small corpus can provide significant benefits Gururangan & et al (2020), and ease of evaluation by a consistent set of reviewers. Each record in the reviews contains a review text, review title, star rating, anonymized ID, and coarse-grained product category, we use only review texts. All reviews were truncated after 2,000 characters, and all reviews were at least 50 characters long. Among the languages present, we used only English for ease of interpreting the results. We used 90%, 5%, and 5% of each data set as training, validation, and test sets, respectively.

**Experiment Setup** We implemented DST by using Pytorch 2.3[5] and will release this code soon. We set $\epsilon$ in Eq (6) to 0.2, and $\lambda_{KDM}$ in Eq (7) to 0.5. As the average length of each text used in fine-tuning the data set is around 60, we set the maximum input sequence length to 64. Note that the ground truth texts were excluded from the training/validation data to prevent information leakage. DST uses GPT-2 medium and large (GPT-2) as the PLM, and, BLOOM[6] and Meta-Llama-3-8B (Llama-3) AI@Meta (2024)[7] as the LLM. Following the training setup, we used Adam with $\beta_1 = 0.9$, and $\beta_2 = 0.999$ was used for optimization, over mini-batches to update parameters; the dropout strategy Srivastava et al. (2014) was adopted for network optimization. The learning rate was 3e-5, with linear warm up over the first 500 steps and linear decay, where we set the dropout rate, the weight decay, and the batch size to 0.1, 0.01, and 256, respectively. We conducted all models on 8 Nvidia Tesla V100 GPUs with 256G memory.

---

[2] https://catalog.ldc.upenn.edu/LDC2008T19

[3] https://huggingface.co/datasets/amazon_reviews_multi

[4] https://huggingface.co/datasets/arxiv_dataset

[5] https://pytorch.org/

[5] https://huggingface.co/transformers/pretrained_models.html

[6] https://huggingface.co/bigscience/bloom

[7] https://github.com/meta-llama/llama3

Table 2: Comparison of topic discovery and text clustering: We evaluate all methods with three topic coherence metrics UCI, UMAss and Intrusion (Int.) and a topic diversity (Div.) metric. We set the number of topics $K = 100$ for all compared methods. Higher score means better for all metrics.

| Methods | discovery | | | | clustering |
|---------|-------|-------|------|------|------------|
| | UMass | UCI | Int. | Div. | |
| BERTopic | -3.76 | -0.50 | 0.71 | 0.62 | 0.27/0.23 |
| TopClus | -2.65 | -0.46 | 0.92 | 0.92 | 0.45/0.27 |
| DST | -2.33 | -0.41 | 0.95 | 0.98 | 0.47/0.28 |

## 5.2 TOPIC DISCOVERY AND TEXT CLASSIFICATION

**Method Baselines:** To evaluate the effect of DST on text classification, a representative task for Transformer encoder framework, BERT, we compare DST with strong BERT-based topic models, BERTopic Grootendorst (2022) and TopClus Meng et al. (2022).

**Evaluation metrics and results:** Following the implementation details and parameters of Grootendorst (2022); Meng et al. (2022), we evaluate the quality of the topic discovery and text classification. Topic discovery involves identifying underlying themes within the data, while text classification involves categorizing text into predefined labels. As good topic results should be both coherent to permit human interpretation and diverse enough to cover more information over the given corpus, we use three metrics including both human (i.e., UMass Mimno et al. (2010), UCI Newman et al. (2010)), automatic evaluations (i.e., Intrusion Meng et al. (2022)), and topic diversity Dieng et al.; Meng et al. (2022), and report model performance under these metrics in Table 2. We conducted text clustering by using $K$-means over the text-level learned latent text embedding, and report the Normalized Mutual Information (NMI) score between the clustering results and the ground truth text labels in Table 2, where we follow the detailed label statistics as found in Meng et al. (2020); the topic label set and location label set are used for the NYT dataset. Examples from the dataset include classifying Amazon reviews into positive or negative sentiments and identifying topics in NYT articles. This table shows that DST achieved an accuracy comparable to TopClus, although it aims to discover differences between linguistic and semantic knowledge and use these differences as topics rather than coherent and meaningful topics.

## 5.3 TEXT GENERATION

**Model Baselines:** As the main application of our framework is to control text generation, we used the latest text generation models with a similar goal as our baselines: fine-tuning model (CO-CON Chan et al. (2021)), prefix-tuning (Prefix Li & Liang (2021) and NRP Carlsson et al. (2022)), and adaptation modules (LoRA Hu et al. (2022), AdaMix Wang et al. (2022), and ReFT Wu et al. (2024)). These experiments use publicly available models[8],[9],[10],[11],[12],[13], and follow the published parameter settings for fair comparison. To evaluate the effect of DST over fine-tuning models, we customized the original tokenizer to extract, as tokens, the top 100 most frequent occurrences of each piece of data not included in the original tokenizer, trained a new representation for each, and evaluated its effectiveness.

**Automated evaluation:** We used test-set perplexity, Dist Li et al. (2016), BLEU-N Papineni et al. (2002), METEOR Lavie & Agarwal (2007), and ROUGE Lin (2004) metrics to measure performance Sai et al. (2023) using the Hugging Face Metrics[14]. $n$-gram based metrics (Dist, BLEU, METEOR, ROUGE) count the overlap between the generated text, and its corresponding reference

---

[8]https://github.com/huggingface/transformers
[9]https://github.com/uber-research/PPLM
[10]https://github.com/xxbidiao/plug-and-blend
[11]https://github.com/alvinchangw/COCON_ICLR2021
[12]https://github.com/FreddeFrallan/Non-Residual-Prompting
[13]https://github.com/XiangLi1999/PrefixTuning
[14]https://huggingface.co/docs/datasets/how_to_metrics

Table 3: Comparison of various PLMs and PEFTs, and ablation analysis: In the model column, (+) means the model with the customized tokenizer. In each model row, the top, and the bottom is the result of Amazon, and arXiv, respectively. As with prefix-tuning models (Prefix and NRP), prefix is a pair of user ID and product ID (Amazon) and each tokenized title (arXiv; avg 11.5). In the column of DST, $K$ and $\mathcal{F}$ denote the number of $\mathbf{z}$ and the kind of transformations, where ad, mu, and af denote addition, multiplication, and affine in Eq (4), respectively. Flu, PPL, D-$N$ and B-$N$ denotes Fluency, Perplexity, Dist-N, and BLEU-$N$, respectively. The bold value denotes the statistical significance for $p < 0.01$ using the Student's t-test, compared to the best baseline.

| Evaluation | | | Human | Automated | | | | Rouge-L | | | $r_{KSL}$ |
| | | | Flu | PPL | D-4 | B-4 | Meteor | P | R | F1 | |
| | DST K | $\mathcal{F}$ | ↑ | ↓ | ↑ | ↑ | ↑ | ↑ | ↑ | ↑ | |
| Model | | | | fine-tuning setting over GPT-2 medium | | | | | | | |
| COCON | | | 3.12 | 15.93 | 10.52 | 13.5 | 21.2 | 0.09 | 0.09 | 0.09 | - |
| | | | 3.35 | 6.88 | 11.34 | 11.2 | 29.1 | 0.24 | 0.23 | 0.23 | - |
| COCON(+) | | | 3.18 | 15.81 | 10.68 | 13.9 | 21.5 | 0.09 | 0.09 | 0.09 | - |
| | | | 3.41 | 6.81 | 11.88 | 11.7 | 29.6 | 0.24 | 0.23 | 0.23 | - |
| PEFT | | | | ablation: DST with GPT-2 medium | | | | | | | |
| | 10 | af | 3.43 | 14.56 | 14.32 | 17.4 | 22.7 | 0.11 | 0.10 | 0.10 | 0.31 |
| | 10 | af | 3.61 | 5.22 | 15.15 | 13.3 | 30.3 | 0.27 | 0.27 | 0.27 | 0.29 |
| DST | 20 | af | 3.63 | 13.03 | 14.22 | 18.2 | 24.8 | 0.14 | 0.14 | 0.14 | 0.32 |
| | 20 | af | 3.69 | 4.82 | 16.12 | 14.0 | 30.2 | 0.28 | 0.28 | 0.28 | 0.31 |
| | 30 | af | 3.66 | 12.92 | 14.34 | 18.3 | 24.9 | 0.14 | 0.15 | 0.14 | 0.32 |
| | 30 | af | 3.72 | 4.80 | 16.19 | 14.1 | 30.8 | 0.28 | 0.28 | 0.28 | 0.31 |
| DST(+) | 10 | af | 3.67 | **12.82** | **14.66** | **18.8** | **25.9** | **0.16** | **0.16** | **0.16** | 0.38 |
| | 10 | af | 3.72 | **4.77** | **16.32** | **14.5** | **31.7** | **0.30** | **0.31** | **0.30** | 0.37 |
| Model | | | | prefix-tuning setting with GPT-2 large frozen | | | | | | | |
| Prefix | | | 2.99 | 16.21 | 10.22 | 14.4 | 20.3 | 0.09 | 0.09 | 0.09 | - |
| | | | 3.21 | 7.12 | 11.18 | 11.2 | 29.2 | 0.21 | 0.22 | 0.21 | - |
| NRP | | | 3.08 | 15.86 | 10.67 | 13.3 | 21.2 | 0.09 | 0.09 | 0.09 | - |
| | | | 3.31 | 7.02 | 11.42 | 11.2 | 30.1 | 0.23 | 0.22 | 0.22 | - |
| PEFT | | | | adaptation modules with GPT-2 large frozen | | | | | | | |
| LoRA | | | 3.02 | 15.72 | 10.72 | 13.8 | 21.6 | 0.10 | 0.10 | 0.10 | - |
| | | | 3.38 | 6.91 | 12.92 | 11.8 | 30.3 | 0.23 | 0.22 | 0.22 | - |
| AdaMix | | | 3.12 | 15.64 | 10.81 | 14.3 | 21.9 | 0.10 | 0.10 | 0.10 | - |
| | | | 3.38 | 6.88 | 12.95 | 11.8 | 30.5 | 0.23 | 0.22 | 0.22 | - |
| ReFT | | | 3.23 | 15.42 | 11.72 | 14.8 | 21.8 | 0.10 | 0.10 | 0.10 | - |
| | | | 3.43 | 6.88 | 12.23 | 12.4 | 30.2 | 0.23 | 0.22 | 0.22 | - |
| | | | | ablation: DST with GPT-2 large frozen | | | | | | | |
| | 10 | ad | 3.41 | 13.86 | 14.42 | 17.1 | 22.6 | 0.12 | 0.12 | 0.12 | 0.28 |
| | 10 | ad | 3.58 | 4.96 | 15.06 | 13.2 | 30.9 | 0.28 | 0.28 | 0.29 | 0.27 |
| DST | 10 | mu | 3.43 | 13.86 | 14.44 | 17.2 | 22.7 | 0.12 | 0.12 | 0.12 | 0.28 |
| | 10 | mu | 3.58 | 4.93 | 15.09 | 13.5 | 31.1 | 0.28 | 0.28 | 0.29 | 0.27 |
| | 10 | af | 3.49 | **13.41** | **14.72** | **17.8** | **23.5** | **0.14** | **0.13** | **0.13** | 0.32 |
| | 10 | af | 3.65 | **4.73** | **15.52** | **14.1** | **32.9** | **0.31** | **0.28** | **0.29** | 0.30 |

Table 4: The contribution of DST (K=10, $\mathcal{F}$=af) to PLMs: Details and meaning are the same as Table 3. The value excluding $r_{KSL}$ is the improvement (+%)

| PLM | Flu | PPL | D-4 | B-4 | Meteor | Rouge-L | | | $r_{KSL}$ |
| | | | | | | P | R | F1 | |
| BLOOM | 6.18 | 13.21 | 10.21 | 12.3 | 11.1 | 10.02 | 10.11 | 10.09 | 0.28 |
| | 6.42 | 13.52 | 10.31 | 12.8 | 11.3 | 10.34 | 10.28 | 10.32 | 0.29 |
| Meta-Llama-3-8B | 5.12 | 10.23 | 8.74 | 9.4 | 9.5 | 8.76 | 8.31 | 8.82 | 0.29 |
| | 4.98 | 9.56 | 8.78 | 9.1 | 9.3 | 8.56 | 8.24 | 8.64 | 0.28 |

Table 5: Case study for Amazon: (top) Ground Truth, (center) AdaMix, and (bottom) DST. We set seed words of "I am disappointed in this purchase", and show the text generated by each model.

| |
|---|
| I am disappointed in this purchase. I bought one of these in another color and in size XL |
| The color is not as vibrant as I would like. It does however still look great. I will use |
| I ordered an XL size in black which arrived with a large hole. There's no way anyone |

text in the test data. We define the metric $r_{KSL}$ to evaluate the effect of KSL;

$$r_{KSL} = \frac{1}{|\mathcal{X}_t|} \sum_{i \in \mathcal{X}_t} \frac{1}{|\mathbf{x}_i|} \sum_{j \in \mathbf{x}_i} z_{ij}, \quad z_{ij} = \begin{cases} 0 & \text{if } z_{ij} = 0 \\ 1 & \text{else,} \end{cases} \quad (8)$$

where $\mathcal{X}_t$ is the set of test text, $\mathbf{x}_i$ is the set of tokens in $i$-th text, and $z_{ij}$ is the topic indicator in Eq (3). The larger this value is, the more knowledge other than "residual" are selected in each token generation, as shown in Eq (4).

**Human evaluation:** We employed fluency testing on attribute relevance as the human annotation Dathathri et al. (2020). Annotators were asked to evaluate the fluency of each sample on a scale of 1-5, with 1 being, 'not fluent at all', and 5 being 'very fluent', as done in Lample et al. (2019). Topic reports the fraction of samples matching the target domain as evaluated by the manual annotators. To consistently evaluate results, we recruited and screened colleagues who were familiar with movies, music (Amazon), and machine learning (arXiv) and could interpret texts.

**Comparisons:** As shown in Table 3, DST outperformed the baselines and achieved better performance over both data sets. These results support our hypothesis that KSL allows DST to distill knowledge in the form of topics, and update only the target-specific token distributions to prevent catastrophic forgetting. Under the fine-tuning setting, KSL emphasizes the target-specific tokens, as shown by the value of $r_{KSL}$, reflects them in the generated texts, and yields improved their quality. Comparing COCON(+) and DST(+), we can say that the tokenizer customization makes it easier for them to extract more target-specific tokens than without it, leads to an increase of $r_{KSL}$, and contributes to improvements in text quality. In the frozen setting (i.e., prefix-tuning and adaptation modules), the target-specific tokens are split into tokens rather than tokens because the tokenizer cannot be customized, but as in the previous setting, we see an improvement in $r_{KSL}$ and a corresponding improvement in text quality. In principle, DST avoids catastrophic forgetting by using the residual in Eq (3) and freezing PLMs. These experiments also show that KSL reflects the target domain, knowledge embedded in the target datasets, in the generated texts, improves their quality, and mitigates this forgetting, as the value of $r_{KSL}$ and the quality of the generated texts are directly related. Table 4 shows that DST contributes the latest s.t.a LLMs (i.e., BLOOM and Llama3).

**Ablation analysis:** We conducted an ablation analysis to investigate the contributions of DST components, specifically $K$ and $\mathcal{F}$. We removed different components and found that the full DST setting achieved better performance across both datasets, as shown in Table 3. The table indicates that there are many settings with values around 0.31. We observed that the knowledge layer extracted target-specific token characteristics for 20% of the total, compared to the case without this layer. Both datasets consisted of review texts, and the ratio of target-specific tokens embedded in PLMs (e.g., GPT-2) is considered to be similar. Although simply increasing the number of knowledge, $\mathbf{z}$, does not dramatically improve $r_{KSL}$ as tokenizer customization does, this value can be further increased by identifying more target-specific tokens and accommodating them in knowledge when combined with this customization. Additionally, the comparison between GPT-2 medium and GPT-2 large indicates that this value appears slightly lower due to the increased number of tokens included in a larger model.

**Error analysis and limitations:** A manual error analysis showed that some instances marked as errors were correctly assessed as allowed by partial matching of tokens in a text. When the ground truth text is personalized, human judgment is hard even if the generated text differs from the ground truth, see Table 5; note that the generated texts contain more abstract or higher frequency tokens than the reference sentences. Our approach could avoid the catastrophic forgetting of linguistic knowledge while not showing any grammatical errors beyond those of other models, especially for the arXiv dataset. One limitation is that it cannot explicitly handle ethical expressions in the given datasets, but this issue will be overcome in future work. While DST maintains the original PLM's

performance, it may struggle when the target data overlaps significantly with the source data of the PLM, leading to unchanged knowledge distributions.

# 6 DISCUSSION AND LIMITATION

DST introduces $\mathbf{z}$ as an indicator for knowledge equivalent subnetworks through KSL and KDM. DST performs domain shift by recalculating $P_\theta(x_{d,t}|z_t, \mathbf{x}_{d,1:t-1})$ and aligning $P_\theta(z_t|\mathbf{x}_{d,1:t-1})$ with the target domain, as shown in Eq (2) and Figure 1. This mechanism ensures that DST differs from the existing topic model and its extensions Wang et al. (2020); Dieng et al.; Chang et al. (2021); Kawamae (2021); Grootendorst (2022); Meng et al. (2022) in understanding the difference between the source (e.g., training corpus) and the target, and in supporting additional learning. Our approach can be applied to both encoder-only (BERTopic) and decoder-only (GPT-2, BLOOM, Llama-3) architectures, demonstrating its versatility and effectiveness, as shown in Tables 2, 3, and 4.

While PEFT methods are known for their data efficiency, DST aims to further improve this by incorporating knowledge and providing additional context and structure to the data. By using this concept to identify and overlay relevant knowledge, DST can enhance the capabilities of PLMs without requiring extensive retraining. Section 6 and Table 3 indicate that DST, like other PEFT methods, allows fine-tuning while keeping the PLM frozen. The additional parameters introduced by DST ($\mathbf{W}_Z \in \mathbb{R}^{d_h \times K}$, $K \times \mathbf{W}_{az} \in \mathbb{R}^{d_h \times d_h}$, and $K \times \mathbf{b}_z \in \mathbb{R}^{d_h}$) result in a total of 5,906,688 parameters for $d_h = 768$ and $K = 10$. This is significantly fewer than the 345 million parameters of GPT-2 medium and comparable to LoRA and other PEFT methods. However, as $K$ increases, the computation time also increases due to the need for calculations for each knowledge, $z$. This overhead is managed by parallel processing for different $z$, as shown in Table 3.

The knowledge in DST is designed to capture subnetworks, similar to how dropout in deep learning helps prevent overfitting by randomly omitting units during training. This approach ensures that the model does not rely too heavily on any single feature, thereby enhancing its generalization capabilities. By treating $\mathbf{z}$ as quantized samples of token distributions, DST can dynamically adjust to different contexts, improving the model's adaptability and performance.

DST excels when there is a significant difference between the source and target domains. When the target data is similar to the source data, the benefits of DST are less pronounced. By aligning knowledge distributions, DST can effectively adapt the model to new domains without catastrophic forgetting. This is particularly useful in low-resource settings where fine-tuning large PLMs on limited data can lead to overfitting. However, in cases where the target domain closely resembles the source domain, the knowledge distributions may not differ significantly, reducing the impact of the Knowledge Steering Layer. Additionally, if the target data is already well-represented in the source data, the benefits of DST's dynamic adjustment may be minimal.

The latent nature of knowledge in DST, while beneficial for capturing complex patterns, poses challenges for interpretability. Future work could explore methods to enhance the interpretability of these latent variables. DST mitigates the domain gap by highlighting the target-specific token distributions through knowledge and updating only these distributions, even if $K$ is small. Methods like variational Bayes and Dirichlet processes can determine the optimal number of $\mathbf{z}$ but are computationally intensive. Therefore, the current study focuses on demonstrating the effectiveness of DST, with automatic determination of $K$ as a future research direction.

# 7 CONCLUSION

The paper proposes a PLM tuning framework, DST, that reflects the target domain knowledge in NLP tasks. Unlike other adapters or PEFT, DST places a lightweight network, KSL, on just the top of PLM and fine-tunes it via KDM. The novelty of DST lies in 1) focusing on the domain gap, 2) representing this gap with subnetwork weights over domains, and 3) guiding PLMs towards the target domain. Experiments showed that both KSL and KDM enable DST to allow PLMs to generate valid texts that well reflect even small target data sets.

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
