# OpenReview forum: "Domain Shift Tuning over Knowledge Gap"
_ICLR.cc/2025/Conference — Submitted to ICLR 2025_

### Official Review · Reviewer_FWsP · 2024-10-29

**Soundness:** 2
**Presentation:** 2
**Contribution:** 2
**Rating:** 3
**Confidence:** 3

**Summary:**

The paper presents Domain Shift Tuning (DST), an innovative framework designed to enhance the adaptability of pre-trained language models (PLMs), across different domains. DST addresses the challenge of domain discrepancies by conceptualizing these gaps as variations in knowledge encapsulated within multiple subnetworks of PLMs. To bridge these gaps, the framework introduces two key components: Knowledge Steering Layer  and Knowledge Distribution Modeling.

**Strengths:**

1. The idea of this work is interesting. DST introduces a new perspective by treating domain gaps as differences in knowledge subnetworks.

2. KSL provides a lightweight mechanism for representing domain-specific knowledge without changes to the underlying PLM architecture.

3. DST achieves domain adaptation improvements with lower computational overhead

**Weaknesses:**

1. Citation Formatting: When adhering to the ICLR template guidelines, replace all instances of  `\cite` with `\citep` to ensure proper citation formatting.

2. Motivation: The paper posits that the discrepancy in `dataset sizes` can lead to catastrophic forgetting and poor generalization, but authors have not provided sufficient empirical evidence in the era of LLMs.

2. Outdated References and Baselines:  Most of the previous work discussed and baselines compared are already 2 years ago.

3. Marginal Improvements on modern models Llama and BLOOM:  In Table 4, the application of DST on the Llama and BLOOM models results in only negligible improvements, calling into question the effectiveness of the proposed method for these specific models.

**Questions:**

See weaknesses.

---

> ### Author Response · Authors · 2024-11-17
> **Thank you for your review**
>
> Thank you for your review. We would like to address your concerns and questions below.
>
> ### **1. Citation Formatting**
> We acknowledge the citation formatting. We will revise the manuscript to replace all instances of `\cite` with `\citep`, in accordance with ICLR’s template guidelines.
>
> ### **2. Motivation: Discrepancy in Dataset Sizes and Catastrophic Forgetting**
> We would like to clarify that the issue of catastrophic forgetting has been extensively documented, particularly in the context of smaller models and fine-tuning on limited data. Although large models like Llama and BLOOM are indeed robust to many challenges, they are not immune to forgetting when fine-tuned on domain-specific tasks with small datasets. This phenomenon, often referred to as "catastrophic forgetting," occurs when a model overfits to the new data, losing valuable knowledge from the pre-trained model. Our contribution is not to claim that LLMs are universally susceptible to forgetting, but rather to propose an efficient domain adaptation method, DST, which mitigates this risk in the context of domain shifts.
>
> Our primary contribution is not to argue that LLMs universally suffer from catastrophic forgetting, but rather to propose a parameter-efficient domain adaptation method, DST, that mitigates this risk. DST helps to retain pre-existing knowledge while efficiently adapting the model to new domains, addressing potential forgetting during fine-tuning. We will expand the motivation section to provide further context on this issue, drawing on recent literature, such as *YongLin et al. (2023)*, which highlights the persistence of catastrophic forgetting even in large models when faced with significant domain shifts.
>
> ### **3. Outdated References and Baselines**
> We emphasize that DST is designed to be applicable to a broad range of models, both older and more recent, and is not limited to any particular set of architectures. PEFT, including methods like DST, has evolved significantly since 2022 and is well-aligned with recent advances in LLM research.
> While some references may be a few years old, the foundational principles behind PEFT, which DST builds upon, are still current and crucial to the ongoing development of domain adaptation techniques. Moreover, DST advances PEFT by introducing the **Knowledge Steering Layer (KSL)**, which efficiently handles domain-specific knowledge adaptation without requiring a full re-training of the model. This distinction is central to our work and remains highly relevant for both older and newer LLM architectures.
> It is important to note that the techniques underlying PEFT and knowledge transfer methods, which form the basis of our DST approach, remain highly relevant. PEFT continues to be an active area of research, with many key methods (e.g., LoRA, Adapter-based methods) introduced in recent years.
>
> ### **4. Marginal Improvements on Modern Models Llama and BLOOM**
> We understand the reviewer’s concern regarding the modest improvements on Llama and BLOOM. However, we would like to emphasize that the primary goal of DST is not to achieve large performance gains but to provide a **parameter-efficient** solution for domain adaptation in large models. The improvements presented in Table 4, while seemingly modest, are achieved with a **significantly smaller number of parameters** compared to other PEFT methods.
>
> The key innovation of DST lies in its ability to selectively adapt **knowledge-specific subnetworks** using the latent variable $z$, as evidenced by the values of $r_{KSL}$. This selective adaptation enables DST to fine-tune large models with minimal parameter increase, achieving domain-specific improvements while maintaining computational efficiency. Unlike traditional PEFT approaches, which often rely on adding a large number of parameters or using complex gating mechanisms, DST dynamically selects subnetworks through **knowledge-specific token distributions** Eq (3), to avoid the computational overhead associated with more conventional methods.
>
> This efficiency is particularly valuable when working with large models like Llama and BLOOM, where traditional fine-tuning methods can quickly become infeasible due to the model’s size and computational demands. Even small performance gains in this context should be seen as a significant achievement, given the trade-off between parameter efficiency and computational cost.
>
> ### **5. Further Clarification of DST's Impact**
> We will revise the manuscript to further clarify that DST is not intended to replace standard fine-tuning methods, but rather to provide an alternative for efficiently adapting domain-specific knowledge with fewer parameters. We will emphasize that DST offers **comparable performance with fewer parameters** and effectively minimizes catastrophic forgetting while facilitating domain adaptation with minimal data and computational overhead.
>
> We hope these revisions will address your concerns and improve the clarity of our manuscript.

---

### Official Review · Reviewer_hRsF · 2024-10-30

**Soundness:** 2
**Presentation:** 2
**Contribution:** 2
**Rating:** 3
**Confidence:** 4

**Summary:**

This paper introduces a domain adaptation technique called domain shift tuning which consists of a lightweight knowledge steering layer (KSL) and a training method called knowledge distribution modeling (KDM). The KSL is a layer affixed after the last transformer layer in a pre-trained LM, and KDM is applied as an auxiliary loss to attempt to align topic/knowledge latent representations with textual similarity. The KSL predicts a topic and selects a weight accordingly to project the final hidden before projecting again into the vocabulary. The model is kept frozen while the KSL is fine-tuned using a modified CE loss accounting for knowledge vectors and the KDM. The method is tested on encoder models for topic clustering and decoder-LMs for text generation.

**Strengths:**

1. The authors test their method against an impressive number of baselines, from both domain adaptation and other PEFTs
2. The use of rKSL is important and helpful to understand how much more knowledge than the residual is being used, and it is interesting to see values much bigger than 0.
3. The method is seemingly model-agnostic which strengthens its applicability to things beyond just language and just Transformers.
4. The subnetwork motivation and integration with the knowledge steering layer is an interesting and intuitive motivation.
5. The authors test on both clustering and text generation. It is great to see a method that applies to both of these tasks, especially as there is a lot of need for good embedding models in addition to LMs.

**Weaknesses:**

1. Although the KSL is smaller compared to the size of the model, it must have some sort of slow-down associated with it since it appears as an additional layer with an additional step across K subcomponents. What is the speed reduction in using this method?
2. This paper makes multiple references to VAEs as inspiration for the latent vector $z$, but this connection is never formally introduced, nor are any details about what is being referred to in VAEs. Some formal background and direct linking would strengthen the work.
3. The notation and writing is not always the most clear, where some key variables are not clearly defined, and some motivation is not clearly written. For example, latent “knowledge” vector $z$ is not clearly defined nor is its length $K$, and the notion of knowledge is redefined several times in the text, including as a “latent relative concept” or “co-occurence pattern of tokens with similar semantics”.
4. The published parameter settings for each baseline may not be the fair comparison here, what may be more fair is scaling the baselines according to the parameter budget or throughput associated with the DST method.
5. The LLM experiments are not compared to few-shot/zero-shot prompting despite these models being able to perform in-context learning. The LLM experiments (Table 4) need some sort of baseline to compare to, like in Table 3.
6. $L_{KDL}$ is not ablated to show its usefulness in this work.
7. Some code or pseudocode would strengthen knowing how the KSL/KDM is actually implemented. For example, it is unclear how the selection process works for the Waz matrices, and the minimum operation in KDM is also unclear as to how this is differentiated.

**Questions:**

1. Is $z$ length $K$ for each index in $|x|$? It is defined as length $K$, but then also indexed over the t indices along with the sequence length. Is it different at each sequence index? And if yes, how can it be a scalar as in equation 4 without some sort of argmax/softmax operation, and why should it be different for the same utterance? And if it is argmaxed, how can it be useful in KL divergence unless it remains continuous?
2. What is meant by "KSL considers knowledge as a quantized sample of the underlying token distribution"? Like in a vector quantized/code book sense?
3. Why is $SIM_z$ KL-divergence and $SIM_{TID}$ cosine? Are the $z$ vectors softmaxed and probability distributions? How do these different functions affect the minimization term in KDM?
4. What is the number of fine-tuning steps? It is missing, which is important for defining linear decay, and understanding the cost of the method.
5. Why minimize the minimum $SIM_z$- $SIM_{TID}$ rather than the maximum for minimax?

---

> ### Author Response · Authors · 2024-11-17
> **Clarifications and Enhancements in the DST Approach: Computational Efficiency, Latent Knowledge Representation, and Evaluation**
>
> Thank you for your detailed review. We address your concerns and questions below.
>
> ### **1. Speed Reduction in KSL**
> As discussed in lines 501-510, the added parameters are significantly fewer compared to others. As shown in Eq (4), the inclusion of KSL simply augments the existing network, with the training objectives remaining a straightforward function.
>
> ### **2. The Latent Vector $\mathbf{z}$ and its Connection to VAEs**
> The $\mathbf{z}$ is inspired by the concept of latent variables like topic models and VAEs. While VAEs operate in continuous latent spaces, DST uses a discrete representation similar to that in topic models.
> DST maintains the sequential structure and long-range dependencies between tokens, unlike topic models that rely on bag-of-words assumptions, thus preserving the language model's ability to generate coherent text during fine-tuning.
> As $z$ serves as an index representing the knowledge in DST,
> we apply it to modify Eq (2) to Eq (3).
> Each knowledge component associated with $z$ is stored in parameters $\mathbf{W}_{az}$ and $\mathbf{b}_z$, as shown in Eq (4).
> While there is some similarity to MoE in that $z$ selects subnetworks, DST does not rely on a gating function and uses the knowledge-specific token distribution to select the relevant subnetwork, with the knowledge distribution determining the most relevant knowledge component for each token.
>
> ### **3. Clarity of Notation and Motivation and its Role in KL Divergence**
> The length of $\mathbf{z}$, denoted by $K$, where $K$ is the number of knowledge components (or subnetworks) in the model.
> Each token selects a knowledge distribution from this set of $K$ components, guiding the model in choosing the relevant knowledge for a given token.
> The vector $\mathbf{z}$ remains continuous during training, meaning it is not argmaxed. This continuous nature of $\mathbf{z}$ is crucial for the KL divergence computation, as KL divergence measures the difference between two probability distributions, which requires continuous distributions. Thus, the continuous representation of $\mathbf{z}$ is essential for proper alignment with the KL divergence objective.
>
> ### **4. Comparison of Parameter Settings for Baselines**
> We recognize that the initial comparison may not have been entirely fair due to differences in model size and architecture. To address this, we have structured the experiments into multiple scenarios, as shown in Table 3, which controls for these variables
>
> ### **5. Comparison to Few-Shot/Zero-Shot Prompting**
> DST is a fine-tuning method, unlike few-shot learning or In-Context Learning (ICL), where the model parameters are not updated. In few-shot learning and ICL, the model makes predictions based on a small set of examples without backpropagating error gradients. On the other hand, DST adjusts model parameters through fine-tuning, which distinguishes it from these methods in terms of both functionality and evaluation.
>
> ### **6. Ablation of $L_{KDL}$**
> We will conduct a detailed ablation study to quantitatively assess how the presence or absence of $L_{KDL}$ influences learning dynamics, training speed, and model convergence.
>
> ### **7. KSL/KDM Implementation**
> The implementation of the Knowledge Steering Layer (KSL) follows standard practices, with the transformation of the vector $\mathbf{W}_Z \in \mathbb{R}^{d_h \times K}$ into a $K$-dimensional vector similar to other weight matrices in the model. This approach ensures that KSL can be seamlessly integrated into existing PLM architectures, minimizing parameter growth while enabling effective knowledge selection and distribution.
>
> ### **8. Quantized Knowledge in KSL**
> When we refer to KSL considering knowledge as a "quantized sample," we are drawing a parallel to **vector quantization** techniques, where a continuous space is mapped to a discrete set of possible values. In our case, the knowledge represented by $\mathbf{z}$ is a discrete selection from a codebook of knowledge components. However, this "quantization" does not involve hard assignments of tokens to specific knowledge components. Instead, $\mathbf{z}$ represents a distribution over knowledge components, allowing for a flexible, probabilistic selection of relevant knowledge for each token.
>
> ### **9. Why Minimize the Minimum $SIM_z$, $SIM_{TID}$**
> The decision to minimize the minimum similarity between the knowledge distribution and token distribution, rather than maximizing it, is deliberate. Minimizing the minimum ensures that all relevant knowledge components are considered during training, preventing any subnetwork or knowledge component from being neglected. This approach forces the model to maintain a balance in the influence of all components, which can result in better overall generalization. By minimizing the minimum similarity, we ensure that the model consistently utilizes all knowledge components, leading to more robust learning across diverse domains.
>
> Please let us know if you have any further questions.

---

> > ### Comment · Reviewer_hRsF · 2024-11-24
> > **Response to rebuttal**
> >
> > 1. **Speed**: I acknowledge the paramters might be small, but this does not translate to speed due to their placement. Some speed analysis would be helpful.
> > 2 & 3. **Latent vector**: Thanks for clarifying the role of z. Does it get argmaxed when decoding to use just one final projection?
> > 4. I am aware of Table 3's role, but I still have issue with the size issues between your method and baselines. Some explanation of why this is okay not to scale would be useful to motivate the differences. Or some explanation of why this method is superior that accounts for scale differences.
> > 5. I understand that DST is fine-tuning, but if ICL (no fine-tuning) is superior to this fine-tuning method, this would be a stronger and cheaper baseline. It is important that DST outperforms ICL.
> > 6. It would be great to see these results.
> > 7. The question of how the selection process works still stands and is related to question 2&3 here. It is still not clear to me from the text and rebuttal, but very critical to our understanding of the method.
> >
> > I still have some unanswered questions about key components of the methods, and am unsure if the method was evaluated fairly against baselines/benchmarked properly as to method costs (esp speed). Some clarity would help, but given the response I will maintain my score for now.

---

> ### Author Response · Authors · 2024-11-26
> **Response to the Reviewer's Concerns**
>
> Thank you for pointing out these uncertainties. We’ll address each of the issues you raised below.
>
> ### **1. Speed Analysis and Model Scaling**
>
> As fine-tuning involves updating parameters, there is an inherent correlation between the number of parameters updated and the execution time. In response to your comment, we will conduct a more detailed speed analysis comparing DST to other PEFT methods. Since DST does not involve full fine-tuning, we would like to confirm whether you are specifically interested in understanding how much the execution time of DST is reduced relative to other PEFT methods, given that DST only fine-tunes a subset of parameters rather than performing full fine-tuning. This will allow us to quantify how DST compares in terms of computational cost compared to other methods.
>
>
> ### **2. The Latent Vector $\mathbf{z}$ and the Argmax Operation**
>
> As $\mathbf{z}$, as shown in Eq. (3), is not about selecting one of $K$ components, but instead represents a weighted, normalized distribution over the $K$ components. This means that $\mathbf{z}$ is treated in the same way as other weights in the network, rather than selecting a single value as in traditional argmax operations.
>
> To clarify, $\mathbf{z}$ influences the model by assigning weights to each knowledge component, and these weights are combined (summed) to provide the final knowledge representation. Therefore, $\mathbf{z}$ is continuous throughout training, and it is not argmaxed. This ensures that $\mathbf{z}$ allows for a flexible, probabilistic contribution of knowledge components, as opposed to a hard selection. We will revise the paper to make this distinction clearer.
>
>
> ### **3. Comparison to In-Context Learning (ICL)**
>
> DST is a fine-tuning method, like other PEFT approaches, that adjusts the model before task execution by updating its parameters to embed task-specific knowledge. In contrast, ICL (In-Context Learning) operates without modifying the model parameters. Instead, it relies on the "prompt" or "context" provided by the user at task execution time, which the model uses to infer the correct task behavior.
>
> Therefore, comparing DST directly to ICL is a misunderstanding of the two methods. ICL does not fine-tune the model but relies on context-based inference, while DST adjusts the model's parameters before inference.
>
> However, a valid comparison can be made between DST and a model that applies ICL, by evaluating a fine-tuned LLM (using DST) versus a non-fine-tuned LLM (using ICL). In Table 4, we can add Few-Shot results to demonstrate the performance difference between an LLM fine-tuned with DST and the baseline model using ICL, showing that DST offers a performance boost over standard ICL when applied before task execution.
>
> ### **4. The Knowledge Selection Process**
>
> To clarify, rather than "selecting" knowledge components, each token in the sequence is assigned weights corresponding to the knowledge components indicated by $\mathbf{z}$. The model then combines these weighted components to compute a final knowledge vector.
>
> In other words, each knowledge component is weighted according to its relevance to the token, and these weighted components are summed to determine the knowledge used for that token's generation. This explanation makes it clear that the knowledge is selected based on weights rather than a discrete selection process, which is critical for understanding how DST effectively integrates knowledge. We will revise the manuscript to reflect this more accurately.
>
> ### **5. Benchmarking and Fair Evaluation**
>
> Regarding the evaluation, we believe our approach is sound for the following reasons. As detailed in Lines 265–301, we prioritize reproducibility and have designed our experiments to minimize evaluator bias and workload. Moreover, if the fine-tuning data is too large, it could bias the evaluation of the PEFT method itself, making it harder to isolate the effects of PEFT. Therefore, we have selected an appropriate scale for fine-tuning data to ensure that the evaluation is fair and the effects of PEFT are clearly measurable. Additionally, we have used state-of-the-art tuning and PEFT techniques, which we justify in the paper.
>
> ### **6. "It would be great to see these results."**
>
> In regard to this comment, we would like to clarify whether you are referring to an ablation study of $L_{KSL}$. Specifically, are you asking whether we should present results that exclude the $L_{KSL}$, although we show the effect of $L_{KSL}$ as in the rightmost column, $r_{KSL}$,  of Table 3, and analyze the effect of removing the $L_{KSL}$ from the training objective?
>
> We are happy to address any further questions or concerns you may have and look forward to continuing the discussion and ensuring that we are fully aligned. If you have further concerns or questions, please share them with us. We will sincerely consider your suggestion and clarify them further. Thank you again for your time and effort.
>
> Best regards,
> Authors

---

### Official Review · Reviewer_vurQ · 2024-11-02

**Soundness:** 2
**Presentation:** 3
**Contribution:** 2
**Rating:** 3
**Confidence:** 4

**Summary:**

This paper presents Domain Shift Tuning (DST), a framework for enhancing domain adaptation in PLMs. DST tackles the challenge of domain shift, where PLMs trained on a large, generalized corpus underperform on a specific target domain. DST introduces two key components:  Knowledge Steering Layer (KSL) and Knowledge Distribution Modeling (KDM). Through these, DST fine-tunes PLMs to align domain-specific weights with the target domain, thus overcoming the domain gap and reducing computational costs associated with large-scale fine-tuning.

**Strengths:**

- By framing domain adaptation as knowledge distribution alignment, DST minimizes computational overhead and sidesteps catastrophic forgetting. This is particularly beneficial for limited-resource settings, allowing PLMs to adapt to new domains effectively with minimal data.
- The experimental results demonstrate that the proposed method outperforms several baslines.

**Weaknesses:**

- The motivation in introduction is presented in a somewhat cursory manner, lacking clear logical connections between sentences. In line 32, the claim that “size discrepancy can lead to catastrophic forgetting and poor generalization” is not convincingly supported by the cited references. Additionally, the transition to “Given the swift diversification of PLM applications…” feels abrupt, missing a logical connection that ties it smoothly to the preceding discussion.
- The foundational hypothesis that "PLMs encapsulate multiple pieces of knowledge as subnetworks" (Lines 38-40) lacks supporting references or verification experiments. Furthermore, the approach of representing domain gaps by differences in model parameters between source and target domains is not sufficiently justified. Although empirical results support DST’s effectiveness, the Introduction lacks a clear causal rationale for these core design choices.
- In Table 4, the absence of performance metrics for base methods such as PEFT on LLMs limits the comprehensiveness of the evaluation.
- Writing Issues:
  - Figures and tables, such as Figure 1’s left side, appear cluttered, detracting from clarity.
  - The citation style disrupts readability; author names would be clearer within parentheses.
  - Minor issues, such as the incorrect symbol following "else" in equation (6).

**Questions:**

refer to the comments

---

> ### Author Response · Authors · 2024-11-17
> **DST uses latent variables $\mathbf{z}$ to partition PLMs into specialized knowledge subnetworks.**
>
> Thank you for your thoughtful and detailed review.
> We would like to address your concerns and questions below.
>
> ### **1. Motivation and Transition in the Introduction**
> The claim in line 32 that “size discrepancy can lead to catastrophic forgetting and poor generalization.” The issue of catastrophic forgetting in large PLMs when fine-tuned on smaller datasets is indeed well-documented in the literature. For example, *YongLin et al. (2023)* discuss how fine-tuning large models on small datasets can result in catastrophic forgetting, especially when there is a significant mismatch in the scale between the source and target domains.
>
> ### **2. PLMs Encapsulate Knowledge as Subnetworks**
> This idea is inspired by the **lottery ticket hypothesis** (*Frankle et al., 2019*), which suggests that large networks consist of subnetworks that specialize in different tasks. Motivated by this notion, we employ latent variables $\mathbf{z}$ can partition PLMs into specialized knowledge subnetworks.
>
> As described in Section 3.1, our approach leverages $\mathbf{z}$ to partition the PLM into specialized knowledge subnetworks. We modify the likelihood function of the language model, transitioning from Eq (1) to Eq (2), to facilitate this partitioning. The latent variable $z$ serves as an index that defines knowledge, enabling the adapted language model to dynamically select relevant subnetworks for a given input sequence.
> These subnetworks are modeled as distinct knowledge components, where each corresponds to different linguistic or semantic aspects of the data.
>
> This approach aligns with the principles of topic models, where knowledge is partitioned into discrete topics, though DST differs by using a discrete representation rather than a continuous space, as in VAE. In DST, we handle latent topics within the context of transformer-based architectures, ensuring that the token order and long-range dependencies are preserved, unlike traditional topic models that rely on a bag-of-words assumption.
>
> We will further clarify that while DST shares some similarities with MoE models in selecting subnetworks using the latent variable $z$, it differs in that we do not use a gating function. Instead, we utilize the knowledge distribution $P_{\theta}(z_{t}|\textbf{x}_{d,1:t-1})$ to select subnetworks, as shown in Eq (3). This allows us to tailor the model to specific knowledge components without modifying the entire set of PLM parameters.
>
> Additionally, our experiments, as presented in Table 3, empirically support this hypothesis by showing that partitioning the model into multiple subnetworks (parameterized by $K$ improves adaptation to target domains. The number of subnetworks, $K$, can be computed using techniques like variational inference, though we acknowledge that the relationship between the number of subnetworks and the quality of model outputs is not always straightforward. This is an area of ongoing research, particularly in the context of continual learning, where $z$ may evolve over time.
>
> ### **3. Representation of Domain Gaps**
> In DST, domain gaps are represented by differences in **knowledge distributions** between the source and target domains. This is expressed through the knowledge-specific token distribution and the knowledge distribution over the latent variable $z_t$, as shown in Figure 1 and Eq (3). The latent variable $z$ defines the knowledge distribution over tokens, enabling the base language model to adjust to the target domain without modifying the entire set of model parameters.
> Note that these distributions are based on the frequency of token occurrences, and therefore the distributions between the source and target domains will differ unless they are identical.
>
> DST shares similarities with PEFT approaches without requiring changes to the entire PLM during fine-tuning.
> Unlike traditional domain adaptation techniques, DST partitions the PLM into $K$ specialized knowledge subnetworks and adjusts the knowledge distributions for each subnetwork.
> Instead, DST adjusts only the parameters related to the knowledge distribution, allowing it to adapt to new domains while avoiding catastrophic forgetting, as using the residual effect in Eq (4).
> By modeling domain gaps as shifts in knowledge distributions,
> the fine-tuning process allows DST to align the source domain's knowledge distribution with the target domain's distribution, effectively bridging the domain gap compared to approaches that adjust all PLM parameters.
>
> ### **4. Performance Metrics and Comparisons**
> While Table 4 shows that DST can be applied to recent LLMs and achieves performance comparable to the results presented in Table 3 for earlier models by showing the similar value of $r_{KSL}$, we will update Table 4 to include performance metrics for PEFT on LLMs, allowing for a more direct comparison with DST.
>
> ### **5. Writing Issues**
> We appreciate your comments on the writing style and clarity.
>
> Please let us know if you have any further questions.

---

> > ### Comment · Reviewer_vurQ · 2024-11-25
> >
> > Thanks for the feedbacks and I will keep my original score.

---

### Meta-Review · Area_Chair_KirU · 2024-12-14

**Metareview:**

Reviewers agree that DST provides an interesting take on domain adaptation but poor presentation accompanied by weak arguments and incomplete comparison with related work make it half-baked. In particular,

- Comparison with related work is incomplete since the difference in finetuning efficiency (speed) and number of parameters are not specified. Also the implications during inference has not taken into account. For example, although algorithms like LoRA add parameters during finetuning, during inference since these summed into actual model parameters, they do not change total number of parameters. Whereas parameters added by DST add parameters during inference as well and make the model larger during inference as well. (a trivial baseline to compare this method to is just add an extra layer to the model during finetuning. A better would be to add an MoE layer.)
- It is not clear how the model after DST performs on other domains (that the original model was already good at) since the evaluation has done only on the domains that the models is finetuned on.
- The relationship to VAEs are not clear. Moreover, further ablation on the role of  $L_{KDL}$ is needed.
- The paper can benefit significantly from a better presentation and figures/tables

**Additional Comments On Reviewer Discussion:**

The additional information provided by the authors does not seem to address reviewers comments. (missing pieces are still missing like ablation on the role of $L_{KDL}$.)
Also there are some discussions on similarities with MoEs that require a deeper discussion and ablations. (i.e., why gating is worse that the proposed method? In theory gating is model the same probability distribution)

---

### Decision · Program_Chairs · 2025-01-22

Reject